# A Unified Approach to Universal Domain Adaptation with Single and Multiple Source Domains

## Abstract

Universal domain adaptation (UniDA) imposes no constraints on the label sets of the source and target domains, aiming to transfer knowledge from source to target domain. Existing works typically target either single-source or multi-source UniDA, rarely both. Naively merging multi-source data into a single source domain may lead to negative transfer and performance degradation. Moreover, since multi-source models are often equipped with modules tailored for multi-source data, they are usually not directly applicable to single-source tasks. These challenges hinder the development of a unified framework. In this paper, we propose a unified model based on multi-modal and uncertainty estimation, termed MUEUDA, to address this issue. First, we incorporate multi-modal information, enabling class-level feature alignment between source and target domains using fine-tuning and prompt learning techniques. Second, we extract class-level image feature prototype from the source domain and progressively update them during training. Finally, we introduce a novel uncertainty estimation method that determines whether an image in the target domain belongs to a known or unknown class through a learnable threshold. Extensive experiments are conducted on both single-source and multi-source benchmarks, and our model achieved state-of-the-art performance. The method demonstrates strong performance across both scenarios, balancing effectiveness and generality. The code is available at
https://github.com/jstree365/MUEUDA.

## 1 Introduce

Domain adaptation (DA) aims to generalize knowledge learned from the source domain to the target domain. Many scholars have researched DA techniques, with applications including image classification(Long et al., 2015), object detection(Hsu et al., 2020; Inoue et al., 2018), and image segmentation(Zhang et al., 2017; Li et al., 2019). In real-world, the labels in the target domain are often unavailable, which is referred to as unsupervised domain adaptation (UDA) (Ganin & Lempitsky, 2015; Long et al., 2016). Although most studies assume that the source and target domains share the same label set, in practice, the label set in the target domain are inaccessible. This implies that, in addition to shared classes, both source and target domains may also contain private classes. This phenomenon has led scholars to investigate universal domain adaptation (UniDA).

Fortunately, some scholars have conducted in-depth research on UniDA. You et al. (2019) have provided a clear definition of UniDA. DANCE (Saito et al., 2020) and DCC (Li et al., 2021) use self-supervised learning based on clustering to distinguish common and private classes. These works are based on single source setting. In this paper, we refer to universal single-source domain adaptation as UniSDA. Some researchers have also studied universal multi-source domain adaptation (UniMDA), such as HyMOS (Bucci et al., 2022) and UMAN (Yin et al., 2022), which align source domains with each other and source and target domains. However, it is important to note that these tasks assume the number of source domains is predetermined.

Different DA are illustrated in Figure 1. Typically, DA can be categorized based on the number of source domains into single-source and multi-source DA. Additionally, DA can be classified based on whether they are UniDA methods into non-universal and universal types. Among these, non-

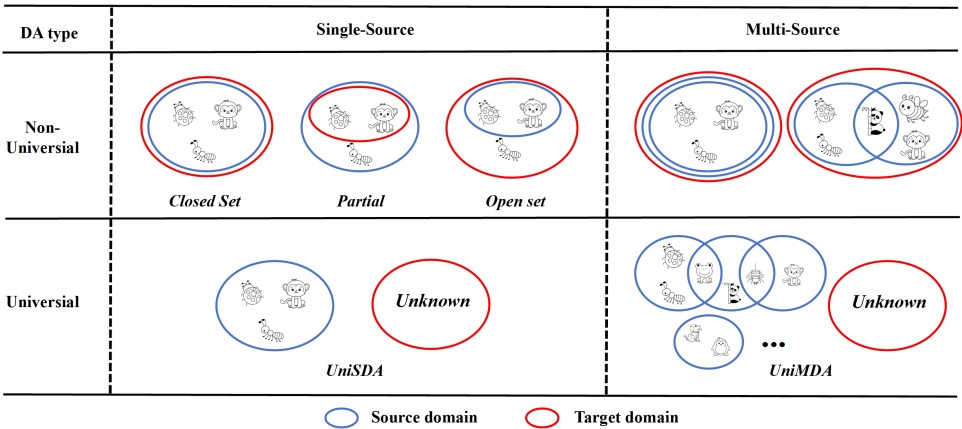

Figure 1: Different DA categories

universal DA also includes closed-set (Long et al., 2015), partial (Zhang et al., 2018; Cao et al., 2018), and open-set (Saito et al., 2018; Luo et al., 2020) scenarios. In this paper, our model addresses both UniSDA and UniMDA to meet practical needs.

When multiple source domains exist, simply merging them into a single source domain ignores inter-source domain discrepancies, leading to potential negative transfer when generalizing to the target domain(Yin et al., 2022). Moreover, UniMDA models are not directly applicable to UniSDA due to their specialized multi-source training designs. These issues motivate us to develop a unified model capable of handling both UniSDA and UniMDA scenarios effectively.

Based on the reasons mentioned above, this paper designs the model based on the following characteristics and requirements: (1) The number of source domains is uncertain; (2) Target domain labels are unavailable; (3) In addition to the shared common labels, both the source domains and the target domain may have private labels; (4) It is necessary to identify the private classes in the target domain. Additionally, current methods are primarily based on image features. Since the source domain data are labeled, these labels are typically used in these methods only to provide supervision information, while the textual information associated with these labels is often ignored. Our model is inspired by this and makes full use of the multi-modal information provided by both text and image, helping align the source and target domains in the class-level feature space.

The contributions of this paper are as follows: (1) We propose the MUEUDA model to address the issue of an uncertain number of source domains in UniDA. To the best of our knowledge, this is the first unified framework designed to handle both single-source and multi-source UniDA, with SOTA results. (2) We introduce multi-modal information into MUEUDA, utilizing the available label information from the source domain. This guides the alignment of the source and target domains in the feature space. (3) We innovatively design an uncertainty estimation method based on class prediction and prototype match similarity to measure the class uncertainty of images, effectively identifying private class samples. This method provides new ideas for open class classification.

## 2 RELATED WORK

### 2.1 UNIVERSAL SINGLE SOURCE DOMAIN ADAPTATION

UniDA is a more generalized form of UDA, which imposes looser restrictions on the label sets of the source and target domain. You et al. (2019) provided a clear definition of UniDA. They employ a domain adversarial training strategy to achieve domain alignment and utilizes uncertainty scores to determine whether an image belongs to an unknown class. Similarly, Fu et al. (2020) leveraged the complementary effects of entropy, consistency, and confidence to more clearly distinguish varying degrees of uncertainty. Chen et al. (2022) designed a cross-domain multi-sample contrastive loss based on mutual nearest neighbors to achieve common class matching and private class separation. However, Saito et al. (2020) argued that such methods ignore the inherent structure of the target

domain. Therefore, DANCE adopted a self-supervised learning approach based on neighborhood clustering to learn the features of known and unknown classes. Similarly, Li et al. (2021) proposed a domain consensus clustering method to mine domain-shared knowledge at both the semantic and sample levels. Zhu et al. (2023) argued that models relying on sample features for judgment overly emphasize global information while neglecting critical local objects in images. They implicitly explored object information in images by sparsely reconstructing attention to achieve better common feature alignment and target class separation. LEAD (Qu et al., 2024) and GLC (Qu et al., 2023) achieved source-free UniDA.

## 2.2 UNIVERSAL MULTI-SOURCE DOMAIN ADAPTATION

UniMDA task must account for domain shifts both between source and target domains and among multiple source domains. HyMOS (Bucci et al., 2022) performs class-balanced alignment between different source domains, and then employs a progressive self-training process to further enhance the alignment between source and target domain clusters. UMAN (Yin et al., 2022) estimates the reliability of each known class belonging to the shared label set by introducing a pseudo boundary vector and its weighted form. These methods are based on visual feature mining and alignment, and they often ignore the textual information provided by source domain labels. SAP-CLIP (Yang et al., 2024a) introduces textual information into UniMDA. It aligns source and target domains at the instance level through image-text alignment. An energy-based uncertainty modeling strategy is proposed to enlarge the margin between known and unknown samples. However, this method relies on a fixed threshold. APNE-CLIP (Yang et al., 2024b) further improves upon this by using a threshold determined by the mean and standard deviation of energy scores to classify whether a sample belongs to an unknown class. To the best of our knowledge, these models are not simultaneously well-suited for both UniSDA and UniMDA. Our method introduces multi-modal information and a novel uncertainty estimation strategy, enabling the model to perform effectively on both single-source and multi-source UniDA, achieving outstanding performance.

## 3 METHODOLOGY

### 3.1 PRELIMINARY

We are given access to $N$ labeled source domains and 1 unlabeled target domain. Let the input space be $\mathcal{X} \subseteq \mathbb{R}^d$ and the label space be $\mathcal{Y} = \{1, 2, \ldots, K\}$. Each source domain is denoted as $\mathcal{D}_s^{(i)} = \{(x_j^{(i)}, y_j^{(i)})\}_{j=1}^{n_i}$, where $x_j^{(i)} \in \mathcal{X}$ is the $j$-th sample from the $i$-th source domain, $y_j^{(i)} \in \mathcal{Y}_i \subseteq \mathcal{Y}$ is the corresponding label, $n_i$ and $\mathcal{Y}_i$ are the number of labeled samples and label set in $i$-th source domain. The unlabeled target domain is represented as $\mathcal{D}_t = \{x_k^t\}_{k=1}^{n_t}$, where $x_k^t \in \mathcal{X}$ and $n_t$ is the number of target domain samples. The corresponding label set of the target domain is denoted by $\mathcal{Y}_T \subseteq \mathcal{Y}$. We define the following sets to characterize label distribution across domains. $\mathcal{Y}_{CS} = \bigcap_{i=1}^{M} \mathcal{Y}_i$, which $\mathcal{Y}_{CS}$ is the **common label set shared across all source domains**. $\mathcal{Y}_S = \bigcup_{i=1}^{N} \mathcal{Y}_i$ is the **total source label set**, $\mathcal{Y}_{T \setminus S} = \mathcal{Y}_T \setminus \mathcal{Y}_S$ is the **target-private label set**, $\mathcal{Y}_{S \setminus T} = \mathcal{Y}_S \setminus \mathcal{Y}_T$ is the **source-private label set**. $\mathcal{Y}_C = \mathcal{Y}_S \cap \mathcal{Y}_T$ is the **shared common label set** between source and target domains. We assume each source domain is drawn from a joint distribution $(x, y) \sim p_i(x, y)$, and the target domain from $(x, y) \sim q(x, y)$, with different marginals $p_i(x)$ and $q(x)$ reflecting domain shifts. The objective of UniDA is to learn a classifier $h : \mathcal{X} \to \mathcal{Y}$ using labeled source data $\{\mathcal{D}_s^{(i)}\}_{i=1}^{N}$ and unlabeled target data $\mathcal{D}_t$, such that $h$ generalizes well to the target domain, especially on the common label set $\mathcal{Y}_C$ and avoids misclassifying target-private samples from $\mathcal{Y}_{T \setminus S}$ as known classes. This paper proposes a model based on multi-modal information and uncertainty estimation (MUEUDA) to achieve alignment between the source domains and target domain, as well as classification of common and private classes in the target domain. The model is applicable to both UniSDA and UniMDA scenarios, and it delivers strong performance without the need for any modifications. The architecture of the proposed network is illustrated in Figure 2.

### 3.2 MULTIMODAL INFORMATION BASED ON TEXT-IMAGE FEATURES

CLIP (Radford et al., 2021), as a large-scale vision-language pre-trained model, possesses strong cross-domain generalization capabilities. However, UniDA tasks require the model to recognize

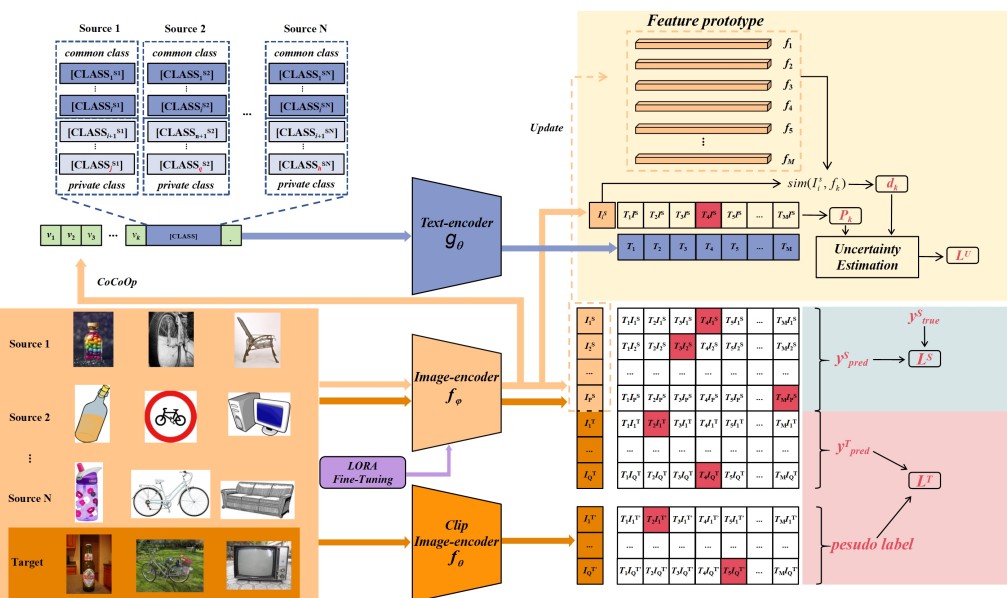

Figure 2: Overview of the proposed MUEUDA approach. Multi-modal information based on image-text is used in the model. Feature prototypes are continuously updated. Pseudo-labeling is used to provide supervision for the target domain samples. We propose a novel uncertainty estimation based on both the feature-prototype similarity and the class prediction confidence to determine whether an image belongs to the common or private classes.

unknown classes, which cannot be accomplished by directly using CLIP alone. There are three core challenges in directly applying CLIP to UniDA: (1) The image encoder, without fine-tuning, may not effectively adapt to the target domain. (2) The static text prompts cannot dynamically adjust to suit the data distribution. (3) CLIP itself lacks the capability to classify unknown classes. To address these challenges, we employ LoRA-based (Hu et al., 2022) fine-tuning to adapt the image encoder, enabling the model to generalize better in target domain. We use instance-based prompt learning to dynamically adjust the text input. Additionally, we construct feature prototypes of $\mathcal{Y}_S$ to assist the model in identifying $\mathcal{Y}_{T \setminus S}$. In the training process, assume there is a set of learnable vectors $[v_1, v_2, \ldots, v_k]$. The source domain image $x_j^{(i)}$ is input into the image encoder to obtain $I_i^S$. Following the approach of CoCoOp (Zhou et al., 2022), we input $I_i^S$ into a Meta-Net to obtain $\pi$ for the image, where $\pi = h_\theta(I_i^S)$. Then the learnable vector $v_k(x)$ is obtained by $v_k(x) = v_k + \pi$. The prompt input corresponding to this image is $t_m(x) = \{v_1(x), v_2(x), \ldots, v_k(x), c_m\}$, where $m \in \{1, 2, \ldots, |\mathcal{Y}_S|\}$ and $c_m$ is the vector corresponding to the label. The text encoder processes the input to obtain $T_m$. $T_m$ and $I_i^S$ are then used to compute the predicted source domain label $y_{pred}^s$ using a softmax over the cosine similarity as follows:

$$p(y = m|x) = \frac{\exp(\cos(I_i^S, T_m)/\tau)}{\sum_{|\mathcal{Y}_S|} \exp(\cos(I_i^S, T_m)/\tau)} \tag{1}$$

The loss $L^S$ is calculated using the predicted labels and the true labels of the image:

$$L^S = -\sum_{i=1}^{M} y_{true}^S \log(y_{pred}^S) \tag{2}$$

To enhance the CLIP image encoder's generalization performance to the target domain, we employ the low-rank adaptation (LoRA) (Hu et al., 2022) to replace its linear layers. Given an input $x \in \mathbb{R}^d$, the output $h \in \mathbb{R}^\alpha$ after adding LoRA is expressed using $\mathbf{W}_0$ and $\Delta W$, $\mathbf{W}_0$ represents the pre-

trained weight and $\Delta W$ is the low-rank approximation of two smaller matrices $\mathbf{B} \in \mathbb{R}^{d \times r}$ and $\mathbf{A} \in \mathbb{R}^{r \times d}$:

$$\mathbf{h} = (\mathbf{W}_0 + \frac{\alpha}{r}\Delta W)x = \mathbf{W}_0 x + \mathbf{B}\mathbf{A}x \tag{3}$$

Here, $\alpha$ is a hyperparameter, and $r$ is the rank of the matrix. The proposed MUEUDA method introduces a teacher network to generate high-quality pseudo-labels for target domain images. During training, the student network uses these pseudo-labels as supervision information to gradually optimize its predictive capability. The teacher network consists of the CLIP image encoder $f_\theta$ and the text encoder $g_\theta$. We obtain pseudo-labels $y_{pesudo}^T$ and predicted labels $y_{pred}^T$ from teacher-net and student-net respectively. The loss $L^T$ is calculated as follows:

$$L^T = -\sum_{i=1}^{M} y_{pesudo}^T \log(y_{pred}^T) \tag{4}$$

### 3.3 Feature Prototype-based Similarity Calculation

Before the training, we input the source domain images into the image encoder to obtain their features. The feature vectors of images belonging to the same class are averaged to obtain the initialized feature prototypes $\{f_1, f_2, f_3, \ldots, f_M\}$, $M = |\mathcal{Y}_S|$. During training, the source domain image with label $j$ is input into $f_\varphi$ to obtain the image feature $I_j^S$, which are then used to update the feature prototypes. The update process is as follows:

$$f_j = \beta f_j + (1 - \beta)I_j^S \tag{5}$$

$\beta$ is a hyperparameter. We use the obtained $I_j^S$ and the output of the text encoder to compute the maximum class prediction probability:

$$P_k = \arg \max_{j \in \{1,2,\ldots,M\}} p(y = j | I_j^S, T_j) \tag{6}$$

We then compute the cosine similarity between $I_j^S$ and the feature prototype $f_k$ corresponding to the class with the highest prediction probability. The result is denoted as $d_k$, calculated as follows:

$$d_k = \text{sim}(I_j^S, f_k)$$
$$\text{sim}(I_j^S, f_k) = \frac{I_j^S \cdot f_k}{\|I_j^S\|\|f_k\|} \tag{7}$$

### 3.4 Uncertainty Estimation

A target sample may fall into one of the following four cases: (1) high $P_k$ and high $d_k$ (ideal known class); (2) high $P_k$ and low $P_k$ (high confidence but semantic shift); (3) low $P_k$ and high $d_k$ (semantically close but low confidence); (4) low $P_k$ and low $d_k$ (typical unknown class). For cases (1) and (2), we argue that high prediction confidence $P_k$ should dominate the decision, meaning that the sample should be considered as belonging to a known class even if its $d_k$ is relatively low. In case (3), high $d_k$ indicates semantic closeness to known classes, so the sample should not be immediately classified as unknown solely due to its low $P_k$. Our goal is to design an uncertainty estimation method that can effectively learn the decision boundary between cases (3) and (4). The proposed formula (8) achieves this by setting the score ranking as $(1) \approx (2) > (3) > (4)$:

$$e^{P_k - d_k} + d_k < \tau \tag{8}$$

Here, $\tau$ is a learnable parameter. In summary, for cases (1) and (2), we use an exponential term to amplify the effect of $P_k$. However, this amplification may cause $d_k$ to be neglected, leading to

samples in case (3) being easily classified as unknown. To address this, we add $-d_k$ as a penalty term to balance the impact of the exponential term. For cases (3) and (4), since the exponential term is close to 1, we linearly add $d_k$ to increase its influence, so that samples with high $d_k$ are more likely to be correctly classified as known classes.

Under mild assumptions, the statistic $S_e(P, d) = e^{P-d} + d$ can be interpreted via a Neyman–Pearson (Lehmann & Romano, 2005) likelihood ratio test, offering some theoretical insight. See Appendix A.1 for details. If the number of samples in a batch that satisfy (8) is greater than or equal to $num = \text{batchsize}/4$, we update $\tau$ using binary cross-entropy loss. First, we compute the prediction logits:

$$y'_k = \tau - (e^{P_k - d_k} + d_k) \tag{9}$$

The true label is set to $y_{\text{unknown}} = 1$, and the BCE loss is calculated as:

$$l^k = \text{BCELoss}(y_{\text{unknown}}, y'_k) \tag{10}$$

The losses $l^k$ for the batch are stored as follows:

$$L = \{l^k | e^{P_k - d_k} + d_k < \tau\} \tag{11}$$

If the number of samples in the batch that meet the condition is $|L|$, then $L^U$ is computed as:

$$L^U = \begin{cases} \frac{1}{|L|} \sum_{l^k \in L} l^k, & \text{if } |L| \geq num \\ 0, & \text{otherwise} \end{cases} \tag{12}$$

During testing, the test image is input into the image encoder $f_\varphi$. $P_k$ and $d_k$ with the corresponding class are then computed. The uncertainty measure $e^{P_k - d_k} + d_k$ is compared with $\tau$. If $e^{P_k - d_k} + d_k < \tau$, the test image is predicted as *unknown*, otherwise, the label is assigned as $k$:

$$label = \begin{cases} \text{unknown}, & \text{if } e^{P_k - d_k} + d_k < \tau \\ k, & \text{otherwise} \end{cases} \tag{13}$$

### 3.5 OPTIMIZATION OBJECTIVE FUNCTION

The overall optimization objective function in this work consists of $L^S$, $L^T$, and $L^U$, expressed as:

$$L_{\text{overall}} = L^S + L^T + L^U \tag{14}$$

During the training phase, We utilize $L^S$ and $L^T$ to update the model parameters, $L^U$ is employed to learn the dynamic threshold $\tau$. In the testing phase, only the student network's image encoder $f_\varphi$, text encoder $g_\theta$, and the updated feature prototypes $\{f_1, f_2, \ldots, f_M\}$ are retained.

### 3.6 THEORETICAL ANALYSIS

Here, We explain why MUEUDA performs so well on UniDA based on Theorem 1, the derived Corollary 1 and Corollary 2.

**Theorem 1.** Assume there exists a fixed feature representation function $\mathcal{Z}_S$ for the source domain and $\mathcal{Z}_T$ for the target domain, such that $\mathcal{Z}_S, \mathcal{Z}_T \in \mathcal{Z}$. Let $\Theta$ be the hypothesis space and $\mathcal{H} \in \Theta$ is a hypothesis subspace. $\epsilon_S$ and $\epsilon_T$ denote the classification errors on the source and target domains, respectively. We define: $\epsilon_S = \sum_{i=1}^{N} \epsilon_{S_i}$. $M$ is the number of source domains. For any classifier $h \in \mathcal{H}$ and an ideal classifier $h' \in \mathcal{H}$, we have:

$$\epsilon_S(h) - \epsilon_T(h, h') \leq \frac{1}{2} d_{\mathcal{H}\Delta\mathcal{H}}(\mathcal{Z}_S, \mathcal{Z}_T) \tag{15}$$

$d_{\mathcal{H}\Delta\mathcal{H}}$ denotes the $\mathcal{H}\Delta\mathcal{H} - distance$. See Appendix A.2 for the proof process

**Corollary 1:** For $\epsilon_T(h)$, based on Theorem 1, there exists a hypothesis space $\mathcal{H}$ with dimension $d$, and $m$ labeled samples drawn from $\mathcal{Z}_S$. Let $\eta' = \epsilon_S(h') + \epsilon_T(h')$, $\hat{\epsilon}_s(h)$ is the empirical estimate for source domain, then with probability at least $1 - \delta$, for every $h \in \mathcal{H}$:

$$\epsilon_T(h) \leq \hat{\epsilon}_S(h) + 4\sqrt{\frac{2em}{d}\log\frac{2em}{d} + \frac{4}{\delta}} + \frac{1}{2}d_{\mathcal{H}\Delta\mathcal{H}}(Z_S, Z_T) + \eta' \tag{16}$$

**Corollary 2:** Let $A(x) = 1(e^{P_k(x)-d_k(x)} + d_k(x) > \tau)$ denote that the target sample $x$ passes the uncertainty filtering, with the probability mass $\sigma = \mathrm{Pr}_{x\sim Z_T}[A(x) = 1]$. Define the filtered target distribution as $\tilde{Z}_T$. Then we have

$$\epsilon_T(h) \leq \hat{\epsilon}_S(h) + 4\sqrt{\frac{2em}{d}\log\frac{2em}{d} + \frac{4}{\delta}} + \eta' + \frac{1}{2}d_{H\Delta H}(Z_S, \tilde{Z}_T) + (1 - \sigma) \tag{17}$$

See Appendix A.2 for the proof process of Corollary 1 and Corollary 2. We observe that $\eta'$ represents the classification error of the ideal classifier on the target and source domains. The main influencing factors on $\epsilon_T(h)$ are the first term $\hat{\epsilon}_S(h)$, which is the empirical training error, the $\mathcal{H}\Delta\mathcal{H}$-distance and $(1-\sigma)$. Specifically, the source domain supervision ($L^S$) is used to minimize $\hat{\epsilon}_S(h)$, multimodal feature alignment techniques are employed to align the source and target domain subsets, thereby reducing $d_{H\Delta H}$. In addition, with $\sigma = \mathrm{Pr}_{x\sim Z_T}[A(x) = 1|(e^{P_k(x)-d_k(x)} + d_k(x) > \tau)]$, this paper leverages effective uncertainty estimation methods to increase $\sigma$, enabling the model to accept more correct known-class samples, which further tightens the upper bound.

## 4 EXPERIMENTS

### 4.1 DATASETS AND EVALUATION PROTOCOLS

**Datasets** Our method is validated on 3 popular domain adaptation datasets. **Office-31** (Saenko et al., 2010) contains 31 classes with 4,110 images across 3 domains: Amazon (A), Dslr (D), and Webcam (W). **Office-Home** (Venkateswara et al., 2017) comprises 65 classes and 15,588 images across 4 domains: Art (A), Clipart (C), Product (P), and Real-World (R). **DomainNet** (Peng et al., 2019) is the largest domain adaptation dataset, 345 classes across 6 domains with approximately 600,000 images total. Following Fu et al. (2020), we select 3 domains (Painting (P), Real (R), and Sketch (S)) from DomainNet for our experiments. We conduct experiments on two UniDA scenarios: UniSDA and UniMDA. Both the source and target domains have private class. For UniSDA, following Chang et al. (2022), we divide the label sets of Office-Home and DomainNet to ensure the fairness of experiments. For UniMDA, we conduct experiments on the Office-31 and Office-Home datasets. We follow the method of Yin et al. (2022); Yang et al. (2024b) to divide the label sets. The specific dataset division methods are described in Appendix A.3.

**Evaluation Protocols** We employ the H-score to evaluate the experimental results, which comprehensively assesses the model's classification performance for both common classes and private classes. The H-score is calculated as:

$$\text{H-score} = \frac{2 \times Acc_{\mathcal{Y}_C} \times Acc_{\mathcal{Y}_{T\setminus S}}}{Acc_{\mathcal{Y}_C} + Acc_{\mathcal{Y}_{T\setminus S}}} \tag{18}$$

where $Acc_{\mathcal{Y}_C}$ represents the classification accuracy for common classes $\mathcal{Y}_C$, $Acc_{\mathcal{Y}_{T\setminus S}}$ denotes the classification accuracy for private classes $\mathcal{Y}_{T\setminus S}$.

### 4.2 EXPERIMENTS DETAILS

Our framework initializes the text and image encoders using a pre-trained CLIP model with ViT-B/16 architecture. Fine-tuning based LoRA method with hyperparameters $r = 8$ and $\alpha = 4$, which are determined based on preliminary experiments. The value of $\beta$ is set to 0.999. We employ

Table 1: Performance Comparison of H-score on Office-31 and Office-Home Datasets for UniMDA

| Protocols | Method | Office-31 | | | | Office-Home | | | | |
|---|---|---|---|---|---|---|---|---|---|---|
| | | 2A | 2D | 2W | Avg | 2A | 2C | 2P | 2R | Avg |
| Source-Combine | CLIP (Radford et al., 2021) | 51.2 | 46.9 | 57.0 | 51.7 | 46.2 | 42.3 | 47.7 | 44.0 | 45.1 |
| | UniOT (Chang et al., 2022) | 45.6 | 38.7 | 36.2 | 40.2 | 34.6 | 42.2 | 41.6 | 37.5 | 39.0 |
| | NCAL (Su et al., 2023) | 52.0 | 48.5 | 57.1 | 52.5 | 45.4 | 40.7 | 28.8 | 39.5 | 38.6 |
| | CMU (Fu et al., 2020) | 72.4 | 74.7 | 71.8 | 73.0 | 77.7 | 61.0 | 64.8 | 71.9 | 68.9 |
| Multi-source | MOSDANET (Rakshit et al., 2020) | 69.2 | 58.8 | 65.4 | 64.5 | 67.1 | 52.1 | 53.7 | 61.5 | 58.6 |
| | TFFN (Li et al., 2023) | 68.6 | 71.6 | 73.4 | 71.2 | 68.9 | 57.4 | 58.7 | 64.1 | 62.3 |
| | HyMOS (Bucci et al., 2022) | 62.3 | 74.9 | 75.3 | 70.8 | 75.7 | 65.8 | 66.3 | 70.8 | 69.7 |
| | UMAN (Yin et al., 2022) | 80.2 | 72.8 | 74.2 | 75.7 | 84.6 | 68.8 | 71.0 | 74.4 | 74.7 |
| | APNE-CLIP (Yang et al., 2024b) | **84.2** | 76.5 | 76.1 | 78.9 | **87.2** | 69.5 | 83.2 | 86.4 | 81.6 |
| | MUEUDA | 82.8 | **83.2** | **83.0** | **83.0** | 86.3 | **78.7** | **88.9** | **90.4** | **86.1** |

Table 2: Performance Comparison of H-score on Office-Home for UniSDA

| | A2C | A2P | A2R | C2A | C2P | C2R | P2A | P2C | P2R | R2A | R2C | R2P | Avg |
|---|---|---|---|---|---|---|---|---|---|---|---|---|---|
| DANN (Ganin et al., 2016) | 42.4 | 48.0 | 48.9 | 45.5 | 46.5 | 48.4 | 45.8 | 42.6 | 48.7 | 47.6 | 42.7 | 47.4 | 46.2 |
| OSBP (Saito et al., 2018) | 39.6 | 45.1 | 46.2 | 45.7 | 45.2 | 46.8 | 45.3 | 40.5 | 45.8 | 45.1 | 41.6 | 46.9 | 44.5 |
| UAN (You et al., 2019) | 51.6 | 51.7 | 54.3 | 61.7 | 57.6 | 61.9 | 50.4 | 47.6 | 61.5 | 62.9 | 52.6 | 65.2 | 56.6 |
| CMU (Fu et al., 2020) | 56.0 | 56.9 | 59.2 | 67.0 | 64.3 | 67.8 | 54.7 | 51.1 | 66.4 | 68.2 | 57.9 | 69.7 | 61.6 |
| DANCE (Saito et al., 2020) | 26.7 | 11.3 | 18.0 | 33.2 | 12.5 | 14.3 | 41.6 | 39.9 | 33.3 | 16.3 | 27.1 | 25.9 | 25.0 |
| DCC (Li et al., 2021) | 58.0 | 54.1 | 58.0 | 74.6 | 70.6 | 77.5 | 64.3 | 73.6 | 75.0 | 81.0 | 75.1 | 80.4 | 70.1 |
| TNT (Chen et al., 2022) | 61.9 | 74.6 | 80.2 | 73.5 | 71.4 | 79.6 | 74.2 | 69.5 | 82.7 | 77.3 | 70.1 | 81.2 | 74.7 |
| UniOT (Chang et al., 2022) | 67.3 | 80.5 | 86.0 | 73.5 | 77.3 | 84.3 | 75.5 | 63.3 | 86.0 | 77.8 | 65.4 | 81.9 | 76.6 |
| OVANet (Saito & Saenko, 2021) | 62.8 | 75.5 | 78.6 | 70.7 | 68.8 | 75.0 | 71.3 | 58.6 | 80.5 | 76.1 | 64.1 | 78.9 | 71.7 |
| GLC (Qu et al., 2023) | 64.3 | 78.2 | 89.8 | 63.1 | 81.7 | 89.1 | 77.6 | 54.2 | 88.9 | 80.7 | 54.2 | 85.9 | 75.7 |
| SAN (Zang et al., 2023) | 68.2 | 80.6 | 86.7 | 73.4 | 73.0 | 79.8 | 76.5 | 64.9 | 83.3 | 80.1 | 67.1 | 80.1 | 76.1 |
| MLNet (Lu et al., 2024) | 68.2 | 83.8 | 85.0 | 73.6 | 78.2 | 82.2 | 75.2 | 64.7 | 85.1 | 78.8 | 69.9 | 83.9 | 77.4 |
| UniAM (Zhu et al., 2023) | 72.0 | 87.1 | **90.7** | 80.3 | 82.4 | 79.8 | 85.0 | 68.4 | 89.0 | 85.4 | 72.1 | 86.1 | 81.7 |
| MUEUDA | **79.0** | **89.2** | 89.7 | **86.2** | **88.5** | **89.8** | **86.8** | **79.0** | **90.5** | **86.9** | **79.3** | **89.9** | **86.2** |

stochastic gradient descent (SGD) optimization with an initial learning rate of $2 \times 10^{-3}$, incorporating a warmup phase (1 epoch, learning rate $2 \times 10^{-5}$) followed by cosine decay scheduling. For prompt tuning, we implement CoCoOp (Zhou et al., 2022) with $N_{ctx} = 4$ learnable context tokens initialized with the template *a photo of a*. The batch sizes is 8. The initial value of $\tau$ is set to 2.1. We conducted our experiments using the PyTorch framework, and all experiments were run on a single GeForce RTX 4090 GPU with 24GB memory.

### 4.3 EXPERIMENTS RESULTS

**Comparison with state-of-the-arts:** To evaluate the performance of our model under the UniSDA and UniMDA settings, we compare it with current SOTA methods. Under the UniMDA setting on the Office-Home dataset, 2A denotes the experiment where Art is used as the target domain. Under the UniSDA setting, A2C denotes the experiment where Art is the source domain and Clipart is the target domain. The best results are shown in **bold**, and the second-best results are underlined. The results for **UniMDA** is reported in Table 1. MUEUDA achieves SOTA results on Office-31 and Office-Home. Specifically, MUEUDA outperforms the previous best method, APNE-CLIP (Yang et al., 2024b), which also a CLIP-based method, by **4.1%** on Office-31 and **4.5%** on Office-Home. For **UniSDA**, as shown in Table 2 and Table 3, our model achieves the best results. Specifically, it surpasses the second-best UniAM (Zhu et al., 2023) by **4.5%** and **7.8%** on the OfficeHome and DomainNet datasets, respectively. Taking the OfficeHome dataset as an example, our model does not exhibit as large a performance gap between the UniMDA and UniSDA settings as UniOT (Chang et al., 2022) and CMU (Fu et al., 2020), achieving balanced and superior performance under both settings. Overall, these results confirm that MUEUDA is highly effective for both UniMDA and UniSDA, delivering superior performance across various benchmarks and settings.

**Analysis of CLIP:** We further investigate whether the superior performance of MUEUDA under UniSDA and UniMDA is solely attributed to CLIP. Based on these two settings, we replace the backbone networks of some methods with the CLIP model for validation on OfficeHome. The experimental results are presented in Table 4. We observe that employing CLIP generally leads to

Table 3: Performance Comparison of H-score on DomainNet for UniSDA

| | P2R | R2P | P2S | S2P | R2S | S2R | Avg |
|---|---|---|---|---|---|---|---|
| DANN (Ganin et al., 2016) | 31.2 | 29.3 | 27.8 | 27.8 | 27.8 | 30.8 | 29.1 |
| OSBP (Saito et al., 2018) | 33.6 | 33.0 | 30.6 | 30.5 | 30.6 | 33.7 | 32.0 |
| UAN (You et al., 2019) | 41.9 | 43.6 | 39.1 | 39.0 | 38.7 | 43.7 | 41.0 |
| CMU (Fu et al., 2020) | 50.8 | 52.2 | 45.1 | 44.8 | 45.6 | 51.0 | 48.3 |
| DCC (Li et al., 2021) | 56.9 | 50.3 | 43.7 | 44.9 | 43.3 | 56.2 | 49.2 |
| OVANet (Saito & Saenko, 2021) | 56.0 | 51.7 | 47.1 | 47.4 | 44.9 | 57.2 | 50.7 |
| SAN (Zang et al., 2023) | 57.8 | 52.9 | 47.9 | 48.4 | 47.2 | 57.9 | 52.0 |
| UniOT (Chang et al., 2022) | 59.3 | 47.8 | 51.8 | 46.8 | 48.3 | 58.3 | 52.1 |
| GLC (Qu et al., 2023) | 63.3 | 50.5 | 54.9 | 50.9 | 49.6 | 61.3 | 55.1 |
| UniAM (Zhu et al., 2023) | 73.9 | 60.9 | 52.3 | 60.0 | 51.4 | 70.7 | 61.5 |
| MUEUDA | **75.9** | **66.5** | **65.6** | **66.3** | **65.4** | **76.3** | **69.3** |

Table 4: Analysis of CLIP on Office-Home for UniSDA and UniMDA settings (H-score)

| **UniSDA** | | **UniMDA** | |
|---|---|---|---|
| Method | Avg | Method | Avg |
| CLIP | 42.1 | CLIP | 45.1 |
| UniOT | 76.6 | UMAN | 74.7 |
| UniOT+CLIP | 78.3 | UMAN+CLIP | 75.6 |
| MLNet | 77.4 | HyMOS | 69.7 |
| MLNet+CLIP | 79.6 | HyMOS+CLIP | 71.2 |
| MUEUDA | **86.2** | MUEUDA | **86.1** |

Table 5: Ablation study of different components on the Office-Home dataset (H-score)

| Setting | UniSDA | | | | | | | | | | | | | UniMDA | | | | |
|---|---|---|---|---|---|---|---|---|---|---|---|---|---|---|---|---|---|---|
| | A2C | A2P | A2R | C2A | C2P | C2R | P2A | P2C | P2R | R2A | R2C | R2P | Avg | 2A | 2C | 2P | 2R | Avg |
| w/o LoRA | 78.7 | 88.9 | 89.4 | 86.0 | 88.2 | 89.6 | 86.5 | 78.9 | 90.3 | 86.7 | **79.3** | 89.9 | 86.0 | 86.3 | 78.5 | 88.4 | 89.8 | 85.8 |
| w/o CoCoOp | 73.0 | 83.5 | 85.0 | 82.3 | 83.3 | 84.9 | 82.7 | 73.1 | 85.4 | 83.7 | 72.7 | 84.0 | 81.1 | 82.4 | 72.1 | 83.3 | 84.9 | 80.7 |
| w/o $L^U$ | 21.9 | 65.8 | 46.1 | 43.5 | 69.4 | 54.6 | 58.4 | 29.2 | 38.1 | 33.3 | 10.3 | 35.1 | 42.1 | 56.0 | 17.0 | 70.5 | 46.0 | 47.4 |
| MUEUDA | **79.0** | **89.2** | **89.7** | **86.2** | **88.5** | **89.8** | **86.8** | **79.0** | **90.5** | **86.9** | **79.3** | **89.9** | **86.2** | **86.3** | **78.7** | **88.9** | **90.4** | **86.1** |

Table 6: Ablation experiments on the OfficeHome dataset (H-score)

| | UniMDA | UniSDA |
|---|---|---|
| $e^{P_k - d_k} + d_k$ | **86.1** | **86.2** |
| $e^{P_k - d_k}$ | 78.7$_{(-7.4)}$ | 77.8$_{(-8.4)}$ |
| $e^{P_k + d_k}$ | 82.2$_{(-3.9)}$ | 81.5$_{(-4.7)}$ |
| $P_k + d_k$ | 81.3$_{(-4.8)}$ | 81.0$_{(-5.2)}$ |
| Only $P_k (\tau = 0.8)$ | 79.8$_{(-6.3)}$ | 80.1$_{(-6.1)}$ |
| Only $d_k (\tau = 0.8)$ | 57.0$_{(-29.1)}$ | 60.8$_{(-25.4)}$ |

performance improvement in the models, but the enhancement is not substantial. This implies that the outstanding performance of MUEUDA is not merely brought by CLIP.

**Components ablation experiment:** To evaluate the impact of LoRA, CoCoOp, and $L^U$ components on the model, we conducted module ablation experiments on Office-Home under both UniSDA and UniMDA settings. The experimental results are presented in Table 5. The results demonstrate that each module contributes positively to model performance, with the best results achieved when all components are included.

**Uncertainty estimation ablation experiment:** We compare against alternative uncertainty metrics, including a simple linear form $P_k + d_k$, two exponential variants $e^{P_k - d_k}$ and $e^{P_k + d_k}$, as well as thresholding-based baselines that rely solely on $P_k$ or $d_k$. The results is shown in Table 6. As observed, our method achieves the best performance.

## 5 CONCLUSION

In this paper, we proposed MUEUDA, a unified model for UniDA that effectively handles both single-source and multi-source scenarios without compromising performance. By introducing CLIP-based multi-modal information, our method leverages fine-tuning and prompt learning to achieve class-level feature alignment between source and target domains. Furthermore, we designed a new uncertainty estimation method to distinguish between the common and private classes in the target domain. This strategy is built upon a combination of model-predicted probabilities and the similarity between features and class prototypes. We conducted extensive experiments and achieve SOTA performance under both the multi-source setting and the single-source setting. This demonstrates that our proposed MUEUDA is a unified UniDA framework with remarkable performance.

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

# A APPENDIX

## A.1 A NEYMAN–PEARSON VIEW OF $e^{(P-d)} + d$ STATISTIC

The following conclusions are derived under idealized distributional assumptions and are intended to help readers understand the related concepts. Let $(P, d) \in [0, 1]^2$ be a sample drawn either from the null distribution $\mathbb{P}_0$ or the alternative $\mathbb{P}_1$. Assume $\mathbb{P}_0$ is uniform on $[0, 1]^2$ with density $f_0(P, d) = 1$, while under $\mathbb{P}_1$ the density is

$$f_1(P, d) \propto \exp\big(-\gamma\, S_e(P, d)\big), \qquad S_e(P, d) := e^{P-d} + d, \ \gamma > 0.$$

For testing $H_0 : (P, d) \sim \mathbb{P}_0$ vs. $H_1 : (P, d) \sim \mathbb{P}_1$, the likelihood ratio satisfies

$$\Lambda(P, d) = \frac{f_1(P, d)}{f_0(P, d)} \propto \exp\big(-\gamma S_e(P, d)\big),$$

which is strictly *decreasing* in $S_e$. Hence, for any fixed type-I error $\alpha$, the most powerful level-$\alpha$ test rejects $H_0$ (i.e., declares "unknown") when

$$S_e(P, d) < t_\alpha,$$

for some threshold $t_\alpha$ chosen such that

$$\mathbb{P}_0\big(S_e(P, d) < t_\alpha\big) = \alpha.$$

By the Neyman–Pearson lemma (Lehmann & Romano, 2005), for simple hypotheses $H_0$ vs. $H_1$ the most powerful test at level $\alpha$ is the likelihood ratio test

$$\phi^*(P, d) = \mathbf{1}\{\Lambda(P, d) > \eta_\alpha\},$$

where $\eta_\alpha$ is chosen so that $\mathbb{P}_0(\Lambda > \eta_\alpha) = \alpha$. In the present construction,

$$\Lambda(P, d) \propto \exp\big(-\gamma S_e(P, d)\big),$$

which is strictly decreasing in $S_e$. Therefore

$$\Lambda(P, d) > \eta_\alpha \iff S_e(P, d) < t_\alpha,$$

with $t_\alpha = -\frac{1}{\gamma} \log \eta_\alpha$. The event $\{\Lambda > \eta_\alpha\}$ has $\mathbb{P}_0$-probability $\alpha$ iff $\{e^{P-d} + d < t_\alpha\}$ has $\mathbb{P}_0$-probability $\alpha$. Consequently, the test $\phi^*(P, d) = \mathbf{1}\{e^{P-d} + d < t_\alpha\}$ is the most powerful level-$\alpha$ test.

Under the above simplified assumptions, the statistic $S_e(P, d) = e^{P-d} + d$ is a monotone function of the likelihood ratio between $\mathbb{P}_1$ and $\mathbb{P}_0$. Hence, thresholding $S_e$ is equivalent to the Neyman–Pearson likelihood ratio test, providing an intuitive statistical interpretation for using $S_e$ as a decision rule.

## A.2 PROOF OF THEOREM 1

**Theorem 1.** Assume there exists a fixed feature representation function $\mathcal{Z}_S$ for the source domain and $\mathcal{Z}_T$ for the target domain, such that $\mathcal{Z}_S, \mathcal{Z}_T \in \mathcal{Z}$. Let $\Theta$ be the hypothesis space and $\mathcal{H} \in \Theta$ a hypothesis subspace. $\epsilon_S$ and $\epsilon_T$ denote the classification errors on the source and target domains, respectively. For multiple source domains, we define:$\epsilon_S = \frac{1}{N} \sum_{i=1}^{N} \epsilon_{S_i}$. For any classifier $h \in \mathcal{H}$ and an ideal classifier $h' \in \mathcal{H}$, we have:

$$\epsilon_S(h, h') - \epsilon_T(h, h') \leq \frac{1}{2} d_{\mathcal{H}\Delta\mathcal{H}}(\mathcal{Z}_S, \mathcal{Z}_T) \tag{19}$$

$d_{\mathcal{H}\Delta\mathcal{H}}$ denotes the $\mathcal{H}\Delta\mathcal{H} - distance$.

**Proof:** From the definition of $\mathcal{H}\Delta\mathcal{H} - distance$, we have:

$$\begin{aligned} d_{\mathcal{H}\Delta\mathcal{H}}(\mathcal{Z}_S, \mathcal{Z}_T) &= 2 \sup_{h,h' \in H} \left| \Pr_{x \sim \mathcal{Z}_S}[h(x) \neq h'(x)] - \Pr_{x \sim \mathcal{Z}_T}[h(x) \neq h'(x)] \right| \\ &= 2 \sup_{h,h' \in H} |\epsilon_S(h, h') - \epsilon_T(h, h')| \geq 2|\epsilon_S(h, h') - \epsilon_T(h, h')| \end{aligned} \tag{20}$$

**Corollary 1:** For $\epsilon_T(h)$, based on Theorem 1, there exists a hypothesis space $\mathcal{H}$ with dimension $d$, and $m$ labeled samples drawn from $\mathcal{Z}_S$. let $\eta' = \epsilon_S(h') + \epsilon_T(h')$, then with probability at least $1 - \delta$, for every $h \in \mathcal{H}$:

$$\epsilon_T(h) \leq \hat{\epsilon}_S(h) + 4\sqrt{\frac{2em}{d} \log \frac{2em}{d} + \frac{4}{\delta}} + \frac{1}{2} d_{\mathcal{H}\Delta\mathcal{H}}(Z_S, Z_T) + \eta' \tag{21}$$

We observe that $\eta'$ represents the classification error of the ideal classifier on the target and source domains. The main influencing factors on $\epsilon_T(h)$ are the first term $\hat{\epsilon}_S(h)$, which is the empirical training error, and the fourth term, the $\mathcal{H}\Delta\mathcal{H}$-distance. Therefore, a good representation should reduce both empirical training error and domain discrepancy.

**Proof:** We have:

$$\begin{aligned} \epsilon_T(h) &\leq \epsilon_T(h') + \epsilon_T(h, h') \\ &\leq \epsilon_T(h') + \epsilon_S(h, h') + |\epsilon_T(h, h') - \epsilon_S(h, h')| \\ &\leq \epsilon_T(h') + \epsilon_S(h, h') + \frac{1}{2} d_{\mathcal{H}\Delta\mathcal{H}}(Z_s, Z_T) \\ &\leq \epsilon_T(h') + \epsilon_S(h) + \epsilon_S(h') + \frac{1}{2} d_{\mathcal{H}\Delta\mathcal{H}}(Z_s, Z_T) \\ &\leq \epsilon_s(h) + \frac{1}{2} d_{\mathcal{H}\Delta\mathcal{H}}(Z_s, Z_T) + \epsilon_T(h') + \epsilon_s(h') \end{aligned} \tag{22}$$

The theorem now follows by a standard application of Vapnik-Chervonenkis (Vapnik, 1999) theory to bound the true error $\epsilon_s(h)$ by its empirical estimate $\hat{\epsilon}_s(h)$. If the source domain provides an i.i.d. sample of size $m$, then with probability at least $1 - \delta$,

$$\epsilon_S(h) \leq \hat{\epsilon}_S(h) + 4\sqrt{\frac{2em}{d} \log \frac{2em}{d} + \frac{4}{\delta}} \tag{23}$$

Plugging this into the previous bound gives,

$$\epsilon_T(h) \leq \hat{\epsilon}_S(h) + 4\sqrt{\frac{2em}{d} \log \frac{2em}{d} + \frac{4}{\delta}} + \frac{1}{2} d_{\mathcal{H}\Delta\mathcal{H}}(Z_S, Z_T) + \eta' \tag{24}$$

**Corollary 2:** Let $A(x) = 1(e^{P_k(x) - d_k(x)} + d_k(x) > \tau)$ denote that the target sample $x$ passes the uncertainty filtering, with the probability mass $\sigma = \Pr_{x \sim Z_T}[A(x) = 1]$. Define the filtered target distribution as $\tilde{Z}_T$. Then we have

$$\epsilon_T(h) \leq \hat{\epsilon}_S(h) + 4\sqrt{\frac{2em}{d}\log\frac{2em}{d} + \frac{4}{\delta}} + \eta' + \frac{1}{2}d_{H\Delta H}(Z_S, \tilde{Z}_T) + (1-\sigma) \tag{25}$$

**Proof:** For any $h, h' \in H$, denote the disagreement set as $S_{h,h'} = \{x : h(x) \neq h'(x)\}$. According to the definition of $H\Delta H$, we need to bound $|\Pr_{x\sim\mathcal{Z}_S}[S_{h,h'}] - \Pr_{x\sim\mathcal{Z}_T}[S_{h,h'}]|$. Since $\tilde{Z}_T$ is the conditional distribution on the event $A(x) = 1(e^{P_k(x)-d_k(x)} + d_k(x) > \tau)$, we can write the decomposition of $Z_T$ as

$$\Pr_{Z_T}[S_{h,h'}] = \Pr_{Z_T}[S_{h,h'} \wedge A] + \Pr_{Z_T}[S_{h,h'} \wedge \neg A] = \sigma\Pr_{\tilde{Z}_T}[S_{h,h'}] + (1-\sigma)\Pr_{Z_T|\neg A}[S_{h,h'}] \tag{26}$$

Hence

$$\Pr_{\tilde{Z}_T}[S_{h,h'}] - \Pr_{Z_T}[S_{h,h'}] = (1-\sigma)\left(\Pr_{\tilde{Z}_T}[S_{h,h'}] - \Pr_{Z_T|\neg A}[S_{h,h'}]\right) \tag{27}$$

Taking absolute values,

$$\left|\Pr_{\tilde{Z}_T}[S_{h,h'}] - \Pr_{Z_T}[S_{h,h'}]\right| \leq (1-\sigma)\cdot 1 = 1-\sigma \tag{28}$$

since the maximum possible difference in probabilities is at most 1. By the triangle inequality,

$$\left|\Pr_{Z_S}[S_{h,h'}] - \Pr_{Z_T}[S_{h,h'}]\right| \leq \left|\Pr_{Z_S}[S_{h,h'}] - \Pr_{\tilde{Z}_T}[S_{h,h'}]\right| + \left|\Pr_{\tilde{Z}_T}[S_{h,h'}] - \Pr_{Z_T}[S_{h,h'}]\right| \tag{29}$$

Then we have,

$$d_{H\Delta H}(Z_S, Z_T) \leq d_{H\Delta H}(Z_S, \tilde{Z}_T) + 2(1-\sigma) \tag{30}$$

Combining Equation (23), we have:

$$\epsilon_T(h) \leq \hat{\epsilon}_S(h) + 4\sqrt{\frac{2em}{d}\log\frac{2em}{d} + \frac{4}{\delta}} + \eta' + \frac{1}{2}d_{H\Delta H}(Z_S, \tilde{Z}_T) + (1-\sigma) \tag{31}$$

### A.3 DATASET SPLITS

#### A.3.1 OFFICE-31

For UniMDA, we adopt the label split approach from Yang et al. (2024b). We select the 10 classes shared between Office-31 and Caltech-256 as common classes, class 1-7 and 4-10 are the common classes for the 2 source domains. The remaining 21 classes are sorted alphabetically. Specifically, the last 10 classes are used as target domain private classes, while the remaining 5 and 6 classes are assigned as source domain private classes for the 2 source domains, respectively.

#### A.3.2 OFFICE-HOME

For UniSDA, we follow the label split approach of You et al. (2019). We use the first 10 classes as common classes, the next 5 classes as source domain private classes, and the remaining classes as target private domain classes.

For UniMDA, we adopt the label division method from Yang et al. (2024b); Yin et al. (2022). The last 50 classes in alphabetical order are treated as target domain private classes. We use the first 10 classes as the common classes, which are assigned alphabetically to the 3 source domains as follows: classes 1–4, 4–7, and 7–10 for each source domain, resulting in 4 common classes per domain. The next 5 classes are used as source domain private classes, which are also assigned alphabetically: classes 1–2, 2–3, and 4–5 to the 3 source domains, with each domain having 2 private classes.

### A.3.3 DOMAINNET

For UniSDA, we follow the label split method of Fu et al. (2020). We use the first 150 classes as common classes, the next 50 classes as source domain private classes, and the remaining classes as target domain private classes. Due to the large dataset size, we select three domains (P, R, and S) for our experiments.

