# OpenReview forum: "A Unified Approach to Universal Domain Adaptation with Single and Multiple Source Domains"
_ICLR.cc/2026/Conference — ICLR 2026 Conference Withdrawn Submission_

### Official Review · Reviewer_GDsJ · 2025-10-22

**Soundness:** 3
**Presentation:** 3
**Contribution:** 2
**Rating:** 4
**Confidence:** 3

**Summary:**

This paper proposes MUEUDA, a unified framework for Universal Domain Adaptation that effectively handles both single-source and multi-source settings. MUEUDA integrates multi-modal features from image–text pairs based on CLIP, enhanced with LoRA fine-tuning and instance-conditioned prompt learning to adaptively align class-level representations between source and target domains. It further maintains progressively updated class prototypes and introduces a novel uncertainty estimation combining class confidence and prototype similarity through a learnable threshold. Extensive experiments on Office-31, Office-Home, and DomainNet datasets show that MUEUDA achieves state-of-the-art performance across both UniSDA and UniMDA benchmarks.

**Strengths:**

* The method defines an uncertainty score $e^{P_k-d_k }+d_k$ with a learnable threshold $\tau$, enabling adaptive separation between known and unknown classes.
* Experiment results demonstrate the effectiveness of the proposed method.
* The paper is well-written and easy to read.

**Weaknesses:**

* The exponential–linear combination $e^{P_k-d_k }+d_k$ lacks clear intuition beyond heuristic discussion; the “penalty” adjustment (lines 270–278) remains empirical.
* Although the authors provide a GitHub link, the repository contains no code at review time; it was created ~5 months ago with empty contents.
* Theoretical results (Theorem 1 and Corollaries 1–2) restate standard domain adaptation bounds without addressing UniMDA-specific challenges (e.g., inter-source divergence).
* Some symbols (e.g.,  $Y_{CS}$ and $Y_C$) are introduced but later used inconsistently; subscript conventions differ between text and equations.
* Figure 2’s legend is dense and difficult to parse; terms like “pseudo-labeling” and “feature-prototype similarity” could be better defined.

**Questions:**

Please refer to Weaknesses part.

---

### Official Review · Reviewer_khwi · 2025-10-26

**Soundness:** 2
**Presentation:** 2
**Contribution:** 2
**Rating:** 2
**Confidence:** 4

**Summary:**

In this paper, the author propose a unified model based on multi-modal and uncertainty estimation to address UniDA, by introducing multi-modal information to enable class-level feature alignment with prompt learning techniques, class-level image feature prototype from the source domain for progressively update, and a novel uncertainty estimation method that determines whether an image in the target domain belongs to a known or unknown class through a learnable threshold.

**Strengths:**

- the idea is simple and easy to follow
- the explanation of threshold design

**Weaknesses:**

__Major Concerns:__
- limited novelty and generality:
  - the theorem is a minor modification of [1]; however, there is no citation which is unprofessional considering this is a submission for top-tier conference.
  - the theorem is less general than [2] considering arbitrary convex combination of source as $\sum_{i=1}^N \alpha_i\epsilon_i$.
  - the theorem only consider open-set risk, while the domain shift caused by source private classes is overlooked, which does not match the motivation; $Z_S$ should be filtered similarly.
  - the domain shift among source domains is not addressed theoretically, but owing to the similarity to the shared encoded prompts, which is strictly limited to VLMs.
  - the tasks of ODA and PDA only are not conducted, making it difficult to assess whether the proposal is a universal solution or not.
- insufficient justification:
  - the case of inconsistent $P_k,d_k$ is ignored; e.g., low $P_k$, low $d_k$ but high $d_{k'}$.
  - the choice of num = batchsize/4 is unexplained.
  - learning curve of $\tau$ is not provided; w/o any constraint, it may becomes very large such that most samples are assigned as unknown.
  - the designed threshold is related to the test to reject null hypothesis that $P,d$ are uniform; however, the guarantee of unknown recognition accuracy is not justified.
  - the performance improvement can be largely attributed to VLMs.
  - the details of the performance regarding $OS^\ast, UNK$ are missing; generally, in open-set problems, there is trade-off between $OS^\ast$\& $UNK$, which should be characterized by $num$.

__Minor Concerns:__
- typos, inconsistent notifications, references before declaration
  - $\mathcal{Y}_{cs}$ is never used
  - undefined $y_{pred}$
  - line 146, $M$ referenced before declaration
  - $M$ denotes differently in line 211 & 236
  - line 196, undefined $k$ (token num?)
  - line 320, missing $1/N$
  - line 258, "high $P_k$ and low $P_k$"

***
[1] A theory of learning from different domains, Machine Learning 2010

[2] Adversarial Multiple Source Domain Adaptation, NeurIPS 2018

**Questions:**

see above

---

### Official Review · Reviewer_YZDA · 2025-10-27

**Soundness:** 3
**Presentation:** 3
**Contribution:** 2
**Rating:** 4
**Confidence:** 4

**Summary:**

The paper presents a coherent and well-motivated approach to unifying UniSDA and UniMDA (Sec. 1; Fig. 1). The integration of multi-modal CLIP features (Sec. 3.2) and the learnable uncertainty estimation (Eq. 8–13) are conceptually sound and empirically validated (Table 6). Theoretical analysis (Sec. 3.6) offers partial insight into the model’s generalization behavior. Experimental evidence across multiple datasets is strong (Table 1–3). However, the clarity and rigor of the theoretical part could be improved (proofs in Appendix A.2 lack precise assumptions). Furthermore, ablations do not isolate the contribution of each modality, and code availability is mentioned but unverifiable (“No direct evidence found in the manuscript”).

**Strengths:**

1. This contributes to the field’s need for generalizable UniDA models.

2. Benchmarks on Office-31, Office-Home, and DomainNet cover diverse adaptation scenario.

3. Ablations and fairness in dataset splits strengthen empirical credibility.

**Weaknesses:**

1. Theorem 1 and Corollary 1–2 (Sec. 3.6; Appendix A.2) restate known $\mathcal{H}\Delta\mathcal{H}$ bounds [a] without domain-specific justification.

2. The proofs omit formal conditions for σ and its relation to τ (Learning criterion in Eq. 12 is heuristic).

3. Code link is cited  in abstract but currently inaccessible (“No direct evidence found in the manuscript”).

4. Critical hyperparameters (e.g., $\tau$ initialization, update frequency, $\beta$ in Eq. 5) are specified but not justified empirically.

5. In Eq. (8), the notation ePk−dk + dk is ambiguous without parentheses; unclear whether d is normalized.

**Questions:**

* Can the authors include t-SNE plots or cosine-similarity heatmaps showing how multi-modal alignment improves class clustering?

---

### Official Review · Reviewer_r4Gm · 2025-10-29

**Soundness:** 2
**Presentation:** 3
**Contribution:** 2
**Rating:** 2
**Confidence:** 4

**Summary:**

In the paper, a unified model based on multi-modal and uncertainty estimation, termed MUEUDA, is proposed to address the development of a unified framework.

**Strengths:**

1.	MUEUDA model is proposed for UniDA.
2.	An uncertainty estimation method is designed.

**Weaknesses:**

1.	The main contribution of the work is that a combination of uncertainty estimation, Multi-source universal domain adaptation, some existed works have been existed, such as [1]. Thus, the contribution and novelty are not enough.
2.	The theoretical analysis has no new theoretical value, which has been used in many works, such as in [1]-[4]. At the same time, no citation for the theorem and corollaries.
3.	The compared methods are all published in 2024. The performance of the proposed method is not persuasive.
4.	The proposed method used CLIP, what is the cost of this utilization? Such as time cost, space occupation, etc.

[1] Style Adaptation and Uncertainty Estimation for Multi-Source Blended-Target Domain Adaptation, In NeurIPS, 2024.
[2] Alexandre Lacoste, and Simon Lacoste-Julien. Pac-bayesian theory meets bayesian inference. In NeurIPS, 2016.
[3] Some pac-bayesian theorems. Mach. Learn., 37(3):355–363, 1999
[4] Analysis of representations for domain adaptation. In NeurIPS, 2007.

**Questions:**

Please see the Weaknesses.

---

### Note · Authors · 2025-12-30

I have read and agree with the venue's withdrawal policy on behalf of myself and my co-authors.